# Clinical Features and Management of Snakebite Envenoming in French Guiana

**DOI:** 10.3390/toxins12100662

**Published:** 2020-10-19

**Authors:** Dabor Resiere, Stéphanie Houcke, Jean Marc Pujo, Claire Mayence, Cyrille Mathien, Flaubert NkontCho, Nicaise Blaise, Magalie Pierre Demar, Didier Hommel, Hatem Kallel

**Affiliations:** 1Intensive Care Unit, Cayenne General Hospital, 97300 Cayenne, French Guiana, France; dabor.resiere@chu-martinique.fr (D.R.); stephanie.houcke@ch-cayenne.fr (S.H.); claire.mayence@ch-cayenne.fr (C.M.); Cyrille.mathien@ch-cayenne.fr (C.M.); didier.hommel@ch-cayenne.fr (D.H.); 2Intensive Care Unit, Martinique University Hospital, 97261 Martinique, France; 3Emergency Department, Cayenne General Hospital, 97300 Cayenne, French Guiana, France; jean.pujo@ch-cayenne.fr; 4Pharmacy Department, Cayenne General Hospital, 97300 Cayenne, French Guiana, France; flaubert.nkontcho@ch-cayenne.fr (F.N.); Nicaise.blaise@ch-cayenne.fr (N.B.); 5Laboratory department, Cayenne General Hospital, 97300 Cayenne, French Guiana, France; magalie.demar@ch-cayenne.fr

**Keywords:** snakebite envenoming, clinical manifestations, French Guiana, *Bothrops atrox*, Antivipmyn Tri^®^

## Abstract

The management of snakebite (SB) envenoming in French Guiana (FG) is based on symptomatic measures and antivenom (AV) administration (Antivipmyn Tri^®^; Instituto Bioclon—Mexico). Our study aimed to assess clinical manifestations, the efficacy, and safety of Antivipmyn Tri^®^ in the management of SB. Our study is a prospective observational work. It was conducted in the Intensive Care Unit (ICU) of Cayenne General Hospital between 1 January 2016 and 31 December 2019. We included all patients hospitalized for SB envenoming. Our study contained three groups (without AV, three vials, and six vials Antivipmyn Tri^®^). During the study period, 133 patients were included. The main clinical symptoms were edema (98.5%), pain (97.7%), systemic hemorrhage (18%), blister (14.3%), and local hemorrhage (14.3%). AV was prescribed for 83 patients (62.3%), and 17 of them (20%) developed early adverse reactions. Biological parameters at admission showed defibrinogenation in 124 cases (93.2%), International Normalized Ratio (INR) > 2 in 104 cases (78.2%), and partial thromboplastin time (PTT) > 1.5 in 74 cases (55.6%). The time from SB to AV was 9:00 (5:22–20:40). The median time from SB to achieve a normal dosage of fibrinogen was 47:00 vs. 25:30, that of Factor II was 24:55 vs. 15:10, that of Factor V was 31:42 vs. 19:42, and that of Factor VIII was 21:30 vs. 10:20 in patients without and with AV, respectively, (*p* < 0.001 for all factors). Patients receiving Antivipmyn Tri^®^ showed a reduction in the time to return to normal clotting tests, as compared to those who did not. We suggest assessing other antivenoms available in the region to compare their efficacy and safety with Antivipmyn Tri^®^ in FG.

## 1. Introduction

Snakebite (SB) envenoming is a frequent event in French Guiana which can cause severe disabilities, or even death [1,2]. Its management requires symptomatic measures and antivenom administration [3]. More than 99 snake species have been identified in French Guiana, of which 12 are venomous, and five of these (*Bothrops atrox, Bothrops brazili, Bothrops bilineatus, Lachesis muta* and *Micruruss* sp) are responsible for most cases of envenomation [4]. Antivenom (Antivipmyn Tri^®^; Instituto Bioclon—Mexico) was introduced in Saint Laurent du Maroni Hospital (Western French Guiana) in 2014 in the frame of a collaboration with the Western France Poison Control Center, to establish a protocol to use an available and efficient antivenom in French Guiana [5]. Antivipmyn Tri^®^ was the sole antivenom against the genus Bothrops authorized for use by the French authorities and available in the national antivenom bank. In January 2017, a 39-year-old man died from severe hemorrhagic complication after being bitten by a snake, presumably Bothrops atrox, in Cayenne (the capital city of French Guiana). At that time there was no available antivenom in Cayenne and snake-bitten patients were managed symptomatically [2]. In 2017, the French Authorities of Health recommended Antivipmyn Tri^®^ for general use at the three main hospitals of French Guiana. Moreover, in September 2017, an international symposium was held at Cayenne under the aegis of the French Regional Health Agency and the Pan American Health Organization [5]. The conclusions of this symposium illustrated the urgent need to ensure the accessibility of effective and safe polyvalent viperid antivenom in French Guiana. In addition, experts stressed the need for post-marketing surveillance to confirm the efficacy of the antivenom used, to define the optimal dosage, and to check for adverse reactions. Common end-points for efficacy were defined as mortality, time taken to restore blood coagulability, halting of hemorrhage, and the importance and reversibility of other clinical manifestations [5]. Our study aimed to assess the clinical manifestations, efficacy, and safety of the antivenom Antivipmyn Tri^®^ in the treatment of snakebites in French Guiana.

## 2. Results

During the study period, 133 patients were admitted to the Intensive Care Unit (ICU) with a diagnosis of snakebite envenoming (on average 44 cases per year). Antivenom was used with 83 patients (62.4%). It was stopped after administration of one or two vials in seven cases (5.2%) because of early adverse reaction. In 76 patients (57.1%), the dose administered was 3 to 6 vials. Of these 76 patients, 14 (18.4%) received 3 vials, 59 (77.6%) received 6 vials, and 3 (3.9%) received 4 to 5 vials. In these three patients, the antivenom administration was stopped because of early adverse reaction. Figure 1 summarizes the distribution of patients according to the antivenom dosage.

The median age of patients was 42 years (28–52) and 70% were male. The median time from snakebite to admission to the ICU was 19:00 (5:03–22:03) and the median time between snakebite and antivenom administration was 9:00 (5:22–20:40). Snake identification was performed in 51% of cases. It was *Bothrops atrox* in 58 patients (43.6%). The main clinical symptoms were edema (98.5%), pain (97.7%), systemic hemorrhage (18%), blister (14.3%), and local hemorrhage (14.3%). The elapsed time from snakebite to the development of systemic hemorrhage was 4.5 h (inter-quartile range (IQR): 0–24). Infection was recorded in 32% of cases. It includes abscess (65%), necrotizing fasciitis (18.6%), and cellulitis (27.9%). Time from snakebite to infection was 6 days (IQR: 3–8). Table 1 summarizes the epidemiological and clinical parameters of our patients.

Symptomatic management was based on analgesics (100%), fluid infusion (92.3%), blood components transfusion (12.8%), dialysis (5.7%), mechanical ventilation (2.3%), and noradrenaline infusion (1.5%). Antivenom was prescribed for 83 patients (62.3%), and 17 of them (20%) developed early adverse reaction. It was mild in 11 cases (13.3% of patients) and severe in 6 cases (7.2% of patients). Antivenom administration was stopped in 10 patients (12%). Fasciotomy was required for 35 patients (26.3%) and necrosectomy was performed on 16 of them (46%). The delay from snakebite to surgery was 7 days (6–9). Table 2 summarizes the management and outcome of our patients.

Biological parameters at admission showed defibrinogenation in 124 cases (93.2%), thrombocytopenia in 52 cases (39.1%), hemolysis in 36 cases (27.1%), rhabdomyolysis in 49 cases (36.8%), and hyper-lactacidemia in 27 cases (20.5%), International Normalized Ratio was >2 in 104 cases (78.2%), and partial thromboplastin time was >1.5 in 74 cases (55.6%). Table 3 summarizes the biologic abnormalities recorded at admission and during a hospital stay.

Overall, the time from SB to ICU was 18:35 (9:33–27:37) in patients without antivenom and 9:00 (4:45–18:10) in those who received antivenom (*p* < 0.001) and time from SB to antivenom was 9:00 (5:22–20:40). Time from SB to normal fibrinogen was 47:00 (28:30–96:13) in patients without antivenom and 25:30 (20:42–32:45) in patients who received antivenom. Time from snakebite to achieve normal dosages of coagulation parameters was shorter in patients who received antivenom than in those who did not (Table 3).

In patients who received antivenom (three or six vials), epidemiologic and clinical parameters were similar (data are shown). However, the Simplified Acute Physiology Score (SAPS) score was 14 (9–19) in the 3 vials group, and 13 (6–17) in the 6 vials group (*p* = 0.017). Furthermore, time from snakebite to ICU was 18:00 (9:30–21:41) in the three vials group and 7:37 (4:19–17:52) in the six vials group (*p* = 0.017). In these patients, biologic parameters at admission and time from snakebite to normal dosages are reported in Table 4. Time from snakebite to achieve normal dosages of coagulation factors was similar in the two groups except for factor VIII (18:00 (11:00–25:15) in the three vials group versus 9:00 (3:51–18:10) in the six vials group; *p* = 0.008), and factor XII (19:15 (10:30–25:20) in the three vials group versus 7:00 (3:00–18:20) in the six vials group; *p* = 0.021). Figure 2 shows the mean and 95% CI of the elapsed time from snakebite to normal dosages of fibrinogen, factor V, and factor VIII according to the number of antivenom vials administered. Table 4 summarizes the biologic abnormalities recorded at admission and during a hospital stay in patients receiving either three or six vials of antivenom.

## 3. Discussion

Our study shows that snakebite envenoming is a public health problem in French Guiana with an average of 44 cases per year hospitalized in Cayenne General Hospital. The median time from snakebite to the hospitalization was 7:30 and to the hospitalization in ICU was 19:00. Kidney injury was observed in 15% of cases and systemic hemorrhage in 18% of cases. The median time for antivenom therapy was 09:00. Local and/or systemic infection was observed in 32% of cases. Two patients died from systemic hemorrhage during the study period. Both of them did not receive antivenom. Patients receiving antivenom therapy showed a shorter time to return to normal coagulation tests, as compared to patients who did not receive antivenom.

Case notification of snakebite is not mandatory in French Guiana and no official territory-level statistics are available. Mutricy and colleagues reported 283 cases of snakebite envenoming who attended the Cayenne Hospital from 2007 to 2015 (8 years) resulting in an average of 31 cases per year and 4 fatal cases for the whole period [1]. This case-fatality rate is among the highest in Latin America, just after Panama, Bolivia, and Guyana [1]. In French Guiana, there are three hospitals (Cayenne, Kourou, and Saint Laurent du Maroni) and 18 health and prevention centers (HPC). Snake bitten patients coming from Cayenne, Kourou, or HPC are managed in the Cayenne Hospital. Thus, our data reflect only the epidemiology of snakebites in these zones of the territory. Further studies investigating all cases attending hospital for snakebite in all of French Guiana are needed.

Snakebite envenoming occurs in most cases among young and healthy persons, predominantly in rural settings, but also in an urban context. Bites occur predominantly in feet and hands and result in local effects and systemic manifestations. Unclottable blood is the most common hemostatic disorder followed by systemic bleeding. Local effects vary from local pain to edema, blistering, hemorrhage, and myonecrosis [6,7,8,9] Systemic manifestations including bleeding, circulatory shock, and acute kidney injury have been strongly associated with fatal cases [1,5]. Additionally, severe local complications may be observed including necrosis, secondary infection and compartment syndrome [2,10,11]. In our study, the main symptoms observed were edema (98.5%), pain (97.7%), local hemorrhage (14.3%), blisters (14.3%), and local necrosis (10.5%). Acute kidney injury occurred in 15% of cases and required dialysis in 50% of them. These proportions are similar to those reported in the literature coming from the Amazon region [6,7,8,9], where most of the cases are inflicted by *Bothrops atrox*, as in French Guiana. Infection was recorded in 32% of cases. The main involved organisms were *Aeromonas hydrophila, Morganella morganii*, and *Bacteroides* sp. These data are in concordance with the published literature [12,13].

### 3.1. Coagulation Disorders

Snake venom induces a consumption coagulopathy including coagulation factor consumption and drops in platelet counts, together with ensuing bleeding disorders. Furthermore, proteolytic enzymes can disrupt the vascular walls, destroy clotting factors and, thus, significantly contribute to the hemorrhagic diathesis. Overall, snake venoms affect almost all components of hemostasis including the vascular wall, platelets, coagulation factors, natural anti-coagulants and fibrinolysis. The mechanism of coagulopathy has been widely described in the literature [14]. Indeed, hypo-fibrinogenemia is a major systemic complication from snakebites, affecting more than 80% of patients [15]. This condition results from the action of serine proteinases having thrombin-like activity, which converts fibrinogen to fibrin, and also to the procoagulant activity of metalloproteinases, which activate factors II and X of the coagulation cascade, resulting in the formation of endogenous thrombin [16]. The action of these venom enzymes causes a consumption coagulopathy. In our study, coagulopathy was recorded in 93% of patients. It was assessed and followed-up by laboratory tests, such as tests for prothrombin time, partial thromboplastin time, fibrinogen dosage, and coagulation factors dosage. Coagulopathy can be assessed using the 20-min whole blood clotting test (WBCT20) [5,17,18]. The WBCT20 can also be used to assess the efficiency of antivenom on coagulopathy [19]. 

The non-coagulability of the blood generally indicates a marked fibrinogen deficiency, with a close relationship between the results of the WBCT20 and the corresponding plasma fibrinogen concentration. Other laboratory tests, such as prothrombin time and partial thromboplastin time, are also used to follow up clotting alterations. In addition, snake venoms induce thrombocytopenia [6] and platelet hypo-aggregation. In our study, these disorders were recorded in most patients. In the whole population, the median time from SB to return to normal fibrinogen concentration was 29:00 (22:18–45:55). It was 25:30 (20:42–32:45) in patients receiving AV and 47:00 (28:30–96:13) in those who did not (*p* < 0.001). It was 33:30 (25:10–44:30) in patients receiving three vials AV and 25:10 (20:33–31:00) in those receiving six vials (*p* = 0.064 when comparing the two antivenom groups). Overall, coagulation tests returned to normal quicker in patients receiving antivenom with no difference recorded between the three-vial and the six-vial groups, except for recovery of factors VIII, IX, and XI. Those results suggest at least a partial efficacy of AV Antivipmyn Tri^®^ to reverse coagulation disorders after SB without a strong difference between the two dosage regimens used. A recent study on the use of the same antivenom in snake-bitten patients attended to in the Guianese western hospital did not reveal differences in the time needed to restore coagulation parameters between patients receiving three vials of antivenom and patients that did not receive antivenom [20]. The reasons for this discrepancy remain to be determined. They could depend on differences between antivenom batches used in the studies or on another as yet unknown factor.

### 3.2. Systemic Bleeding

Systemic bleeding is one of the hallmarks of viperid snakebite envenomation, with unclottable blood and thrombocytopenia as its associated major risk factors [6]. It commonly arises in the case of microvascular damage by proteolytic degradation of the basement membrane [21]. The involved enzymes are PI and P-III snake venom metalloproteinases (examples in the venom of *B. atrox* are Batroxase [22], Atroxlysin-Ia [23] and Batroxrhagin [24]). Systemic bleeding observed in snakebites was reported in 3.6–15.3% of patients [15,25]. It includes gingival bleeding, sub-conjunctival hemorrhage, hematuria, and, in severe cases, cerebral hemorrhage. Unclottable blood and thrombocytopenia on admission were independently associated with systemic bleeding during hospitalization [6]. In our study, systemic bleeding was recorded in 18% of cases independently of the severity of coagulation disorders. It required blood components transfusion in 12.8% of cases and resulted in two deaths.

### 3.3. Antivenom

Several antivenoms are manufactured in Central and South America for the treatment of envenomings by viperid snakes. They include bothropic, crotalic, lachetic, bothropic–crotalic, bothropic–lachetic and bothropic–crotalic–lachetic antivenoms. Some of these antivenoms are made of fragment antigen binding F(ab’)2 antibody fragments, whereas others consist of whole IgG molecules [26]. Preclinical studies have underscored the efficacy of various antivenoms, produced using different mixtures of viperid venoms, for neutralizing the venoms of *B. atrox* and of other Amazonian species [27,28]. Moreover, the efficacy of the bothropic and bothropic–lachetic Brazilian antivenoms, as well as other bothropic and polyvalent antivenoms manufactured in Ecuador and Colombia, have been demonstrated in clinical trials in envenomings by *B. atrox* in the Amazon region [17,25,29]. However, no French manufacturer is producing antivenom for snakes in French Guiana, and no authorization of commercialization according to European law exists for the available antivenoms. For this, the use of antivenom in French Guiana needs a Temporary Use Authorization (TUA) delivered by the National Agency of Drugs. The antivenom recommended and temporarily authorized in French Guiana is Antivipmyn Tri^®^ (F(ab’)2 polyvalent antivenom produced by Instituto Bioclon, in Mexico). In our study, Antivipmyn Tri^®^ was used in patients attending the hospital following snakebite and presenting coagulation disorders. However, there were periods of supply disruption, and patients did not receive antivenom during these periods. In addition, after an intermediate analysis of patients who received three vials per administration, and according to the package leaflet of the product, the dose administered was increased to six vials per administration. Consequently, our study contains three groups (without antivenom, three vials of antivenom, and six vials of antivenom).

### 3.4. Adverse Reaction to Antivenom

In the Brazilian Amazon, early adverse events after antivenom therapy were observed in 19.8% of the patients. The most common were urticaria (13.8%), pruritus (11.2%), facial flushing (3.4%), and vomiting (3.4%). All adverse events were mild and all signs and symptoms of early adverse reactions ceased within 48 h from management [30]. In our study, early adverse reactions were recorded in 17 cases (20.5%) and resulted in stopping antivenom administration in 10 cases. Continuing administration of antivenom in patients with adverse reactions is debated. Moreover, in our study, seven patients received the complete prescribed dose of antivenom despite adverse reactions, with favorable outcomes. Regarding the small size of this group, any conclusion about the administration of antivenom in patients developing adverse reactions would be speculative.

However, we recently studied the preclinical efficacy of freeze-dried antivenoms manufactured in Costa Rica (Polival-ICP^®^) and Mexico (Antivipmyn Tri^®^) against the lethal, hemorrhagic, in vitro coagulant, and myotoxic effects of *Bothrops atrox* venom from French Guiana [31]. These antivenoms differ in protein concentration, and in the type of active principle (IgG and F(ab’)2, respectively). Polival-ICP^®^ showed significantly higher neutralizing activity against lethal, hemorrhagic and in-vitro coagulant activities of the venom. In the case of lethal activity, Antivipmyn Tri^®^ did not neutralize the effect at the highest antivenom level tested (1 mg venom/mL antivenom). A previous study indicated that the dose of three vials is insufficient, especially in grade 2 or 3 cases, since no improvement in the correction of coagulation parameters was observed [20].

Our study has two major limitations. First, this is a mono-centric observational study. However, the Cayenne general hospital provides care for more than 2/3 of the Guianese population. Second, for coagulation tests, no antivenom was added to blood samples to neutralize snake venom present in the blood. Third, there was no dosage to identify the venom type (differentiating Bothrops from Lachesis venoms) and to quantify venom and antivenom in the serum sample. Finally, the snake identification was not very precise because the snake was not seen in many cases and is known by different common names according to the geographic region [32,33].

## 4. Conclusions

Snakebite envenoming is a public health problem in French Guiana and can be responsible for severe envenoming scenarios, including acute kidney injury, systemic hemorrhage, local necrosis, and death. Patients receiving Antivipmyn Tri^®^ showed a reduction in the time to return to normal clotting tests, as compared to those who did not. Our results suggest that Antivipmyn Tri^®^ is effective in the reversion of coagulation disorders in patients with SB envenoming. However, we observed a significant rate of adverse reactions. We suggest assessing other antivenoms available in the region to compare their efficacy and safety with Antivipmyn Tri^®^ in French Guiana. 

## 5. Materials and Methods

Our study is a prospective observational non-interventional work. It was conducted in the Intensive Care Unit (ICU) of Cayenne General Hospital between 1 January 2016 and, 31 December 2019. We included all patients hospitalized for snakebite envenoming regardless of the grade of envenoming. According to the protocol for managing snakebite envenoming in French Guiana, all patients should be attended in the ICU (see below).

Cayenne General Hospital is a 742-bed general center that provides first-line medical care for an urban population of 150,000 inhabitants. It manages 18 de-localized prevention and care centers providing care for almost 50,000 additional inhabitants. Consequently, it is also a referral center for a larger population coming from all over French Guiana and the neighboring countries. In French Guiana there are three emergency units in the three major cities (Cayenne, Kourou, and Saint Laurent du Maroni).

French Guiana is an overseas region of France located on the North Atlantic coast of South America between three- and five-degrees north latitude. The population was estimated at 296,700 inhabitants in 2018, which represents 3.5 inhabitants/km^2^. Inhabitants live mostly on the coast (86%) or along the rivers. Fifty-two percent live in the metropolitan area of the main city of Cayenne, with a density of 574 inhabitants/km^2^. French Guiana is home to unique and important ecosystems: tropical rainforests (96% of the region), coastal mangroves, savannahs, inselbergs, and many types of wetlands [34]. It has a tropical climate, with a distinct dry period and a long rainy season. The rainy season lasts from December to June. This season includes a small rainy season in December and January, followed by a small dry season in February and March and finally the main rainy season from April to June, with a pluviometry peak in May. Daytime temperatures are higher in the forest than on the coast, but nights are cooler. Humidity is high in the whole territory. Humidity and temperatures are relatively constant, being 85% and 27 °C, respectively [35].

The regional protocol for the management of snake-bitten cases stipulates that patients presenting to the emergency department of the three major cities should receive 3 vials of Antivipmyn Tri^®^ antivenom and should be hospitalized in the ICU (clinical toxicology sector). Antivipmyn Tri^®^ is an F(ab’)2 polyvalent antivenom produced by Instituto Bioclon, in Mexico—Mexico Registry N° 58583 SSA IV. According to the manufacturer, it is indicated for the treatment of envenoming by vipers, such as *Bothrops atrox, Bothrops brazili, Bothrops asper, Bothrops neuwiedii, Bothrops alternatus, Bothrops jararacussu, Bothrops venezuelensis, Bothrops pictus, Crotalus durissus terrificus, Crotalus durissus durissus, Lachesis muta stenophrys, Lachesis muta muta, Sistrurus* spp., and *Agkistrodon* spp. Unfortunately, we have had periods of supply disruption when patients did not receive antivenom. After an intermediate analysis of patients who received 3 vials per administration, and according to the package leaflet of the product, the dose administered was increased to 6 vials per administration. Then, our study contained three groups (without antivenom, 3 vials of antivenom, and 6 vials of antivenom).

At admission to ICU, blood tests include hemostasis tests, tests for renal function, creatine kinase (CK) dosage, and all other routine tests performed in ICU. Hemostasis tests include a fibrinogen dosage, International Normalized Ratio (INR), partial thromboplastin time (PTT), as well as the dosage of the coagulation factors (II, VII, VIII, IX, X, XI, and XII). Renal function tests include serum urea nitrogen and creatinine dosage. These blood tests were performed every 6 h from admission until the return of the hemostasis to normality, then daily until discharge from ICU.

In all envenomed patients, we prospectively collected epidemiological and clinical data, including age and sex of patients, the date and time of the bite, the anatomical site of the bite, the snake description, the grade of envenoming, the clinical manifestations at admission and during the hospital stay, and the adverse reactions to antivenom. 

Our study is a prospective observational non-interventional work and did not require individual consent according to the French law regarding research conforming to the norm MR-003 (JORF no. 0160 du 13 juillet 2018. texte no. 109). The protocol of antivenom administration and blood test dosages was approved by the hospital’s institutional review board (Ref: UF3700/17’, version “b”). All patients were informed about the hospital protocol on the management of snakebite and were informed that the data collected will be used in research programs. Verbal consent was obtained from all patients or relatives (when patients are <18 years or unable to consent) and was reported in medical file of the patient by the doctor in charge. The control group was available because we have had periods of supply disruption and patients did not receive antivenom during these periods. So they were not selected nor randomized. In France, the exceptional use or compassionate use of a drug that do not benefit from a marketing authorization (MA) is subject to the prior obtaining of a Temporary Authorization for Use (TAU). When the antivenom (AntivipmynTri®) is administered, a completed form (Blank form reference: Q11ADOC025 v01) including the patient data, the dosage and the route of administration used was completed and returned to the French National Agency for Drug Safety (ANSM: Agence Nationale de Sécurité de Médicaments). In this form, the physician in charge certify that the patient is informed about the drug use and that he undertakes to inform the ANSM of any adverse effect. Our database has been registered at the Commission National de l’Informatique et des Libertés (registration n° 2217025v0), in compliance with French law on electronic data sources.

### 5.1. Definitions

Body Mass Index (BMI) is a person’s weight in kilograms divided by the square of height in meters. The Simplified Acute Physiology Score (SAPS II) is a severity score and mortality estimation calculated on admission to ICU, based on the worst parameters recorded during the first 24 h of hospitalization [36]. Thrombocytopenia is defined by a platelet count < 150 G/L. Defibrinogenation is defined by a fibrinogen level <1 g/L (normal value: 2–4 g/L). Rhabdomyolysis is defined by a CK level > 500 IU/L (normal value: 39–308 U/L). Coagulation disorders are defined by International Normalized Ratio >2 (normal value: 0.8–1.2), Partial thromboplastin time > 1.5, Prothrombin time and coagulation factors < 60%. Renal failure is defined according to the Kidney Disease Improving Global Outcomes (KDIGO) definition [37]. The grade of envenoming is assessed according to the conclusions of the international symposium held in French Guiana in 2017 [5]. Adverse reactions to antivenom were reported to the French Agency for the Safety of Health Products (ANSM: Agence Nationale de Sécurité de Médicaments) and were classified as ‘mild’ (only cutaneous-urticaria, pruritus) or ‘severe’ (bronchospasm, angioedema, hypotension, colic) [38].

### 5.2. Statistical Analysis

We created a data file with the patient’s and snake’s information and we performed a descriptive analysis using Excel (2007) and IBM SPSS Statistics for Windows, version 24 (IBM Corp., Armonk, NY, USA). Results are reported as the median and inter-quartile range (IQR), mean ± standard deviation (95%), or numbers with percentages. Time is expressed as hours and minutes (hh:mm). To compare qualitative variables, we used the Fisher exact test. To compare quantitative variables, we used the student’s t-test or the Mann–Whitney U-test. The significance level was set at *p* ≤ 0.05.

## Figures and Tables

**Figure 1 toxins-12-00662-f001:**
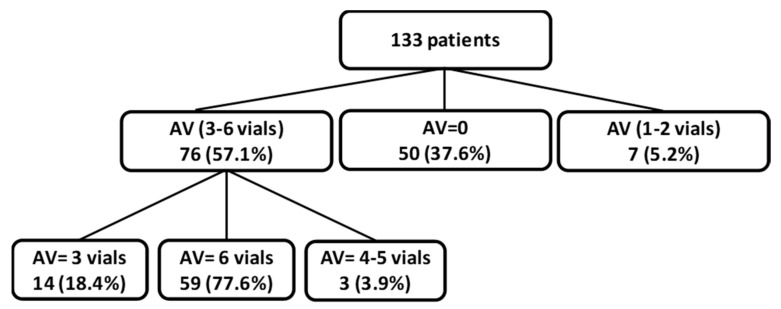
Flowchart showing the distribution of patients according to antivenom dosage. (AV: Antivenom).

**Figure 2 toxins-12-00662-f002:**
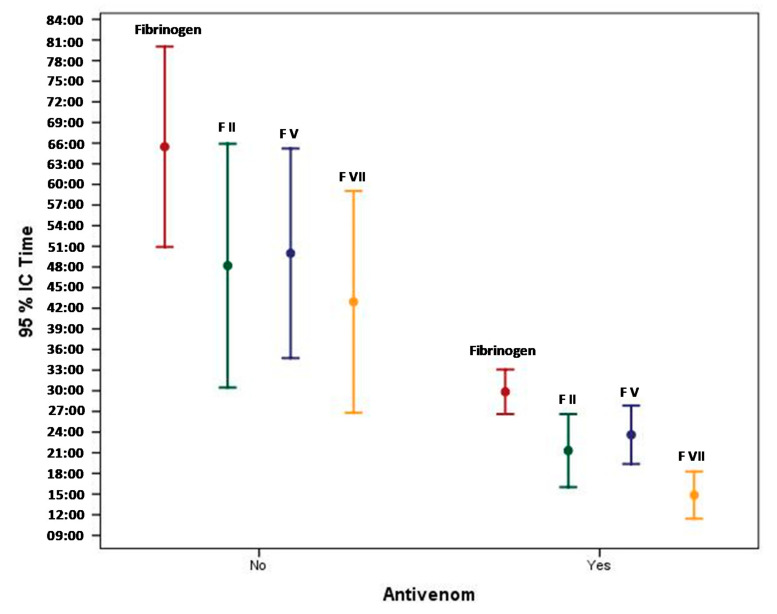
The mean and 95% CI of the elapsed time from snakebite to the time of normal dosages of fibrinogen, factor II, factor V, and factor VIII according to the antivenom use. *p* < 0.001 for all factors. (results are expressed in means ± SD).

**Table 1 toxins-12-00662-t001:** Epidemiological and clinical parameters of our patients.

	All Patients		Without Antivenom		With Antivenom		*p*
Parameter	Nb	Result	Nb	Result	Nb	Result	
Age, years	133	42 (28–52)	50	47 (28–53)	76	40 (29–52)	0.595
Male gender	133	93 (69.9%)	50	37 (74%)	76	52 (68.4%)	0.501
BMI, Kg/m2	117	24 (21–27)	48	24 (22–27)	62	24 (21–27)	0.866
SAPS	133	13 (6–18)	50	13 (8–19)	76	13 (6–18)	0.127
History	133	40 (30.1%)	50	14 (28%)	76	24 (31.6%)	0.668
Hypertension	133	11 (8.3%)	50	2 (4%)	76	8 (10.5%)	0.185
Alcohol abuse	133	6 (4.5%)	50	4 (8%)	76	2 (2.6%)	0.166
Duration							
Time from SB to Hospitalization	133	7:30 (2:00–11:00)	50	11:55 (5:00–27:30)	76	5:12 (1:08–18:05)	0.000
Time from Hospitalization to ICU	128	1:39 (0:00–6:22)	50	3:15 (0:00–6:04)	71	1:30 (0:00–3:55)	0.020
Time from SB to ICU	128	19:00 (5:03–22:03)	50	18:35 (9:33–27:37)	71	9:00 (4:45–18:10)	0.000
Snake Bite							
Grade I envenomation	133	51 (38.3%)	50	14 (28%)	76	32 (42.1%)	0.108a
Grade II envenomation	133	47 (35.3%)	50	18 (36%)	76	28 (36.8%)	0.381b
Grade III envenomation	133	35 (26.2%)	50	18 (36%)	76	16 (21.1%)	0.064c
Clinical features							
Snake identification	133	68 (51.1%)	50	24 (48%)	76	39 (51.3%)	0.716
Oedema	133	131 (98.5%)	50	49 (98%)	76	75 (98.7%)	0.764
Nb of involved segments	133	2 (1–3)	50	2 (2–3)	76	2 (1–2)	0.091
Local hemorrhage	133	19 (14.3%)	50	5 (10%)	76	14 (18.4%)	0.196
Necrosis	133	14 (10.5%)	50	5 (10%)	76	9 (11.8%)	0.748
Blister	133	19 (14.3%)	50	15 (30%)	76	4 (5.3%)	0.000
Pain	133	130 (97.7%)	50	47 (94%)	76	76 (100%)	0.031
Mean arterial pressure, mmHg	133	95 (88–102)	50	93 (86–103)	76	95 (89–102)	0.578
Cardiac rythm, beat/min	133	78 (67–91)	50	84 (72–94)	76	76 (66–90)	0.213
Temperature (°C)	133	37 (37–37)	50	37 (37–38)	76	37 (37–37)	0.001
SpO_2_ (%)	133	99 (98–100)	50	99 (97–100)	76	100 (98–100)	0.274
Shock	133	2 (1.5%)	50	2 (4%)	76	0 (0%)	0.079
Renal failure	133	20 (15%)	50	10 (20%)	76	10 (13.2%)	0.304
Systemic hemorrhage	133	24 (18%)	50	12 (24%)	76	11 (14.5%)	0.176

BMI: Body Mass Index; SAPS: Simplified Acute Physiologic Score; SpO_2_: peripheral capillary oxygen saturation; a: Grade I compared to Grade II and III; b: Grade II compared to Grade I and III; c: Grade III compared to Grade I and II.

**Table 2 toxins-12-00662-t002:** Management and outcome of our patients.

	All Patients		Without Antivenom		With Antivenom		*p*
Parameter	Nb	Result	Nb	Result	Nb	Result	
Dialysis	20	10 (50%)	10	6 (60%)	10	4 (40%)	0.371
Cathecholamines	133	2 (1.5%)	50	2 (4%)	76	0 (0%)	0.079
Mechanical Ventilation (MV)	133	3 (2.3%)	50	3 (6%)	76	0 (0%)	0.031
Duration of MV, days	3	16 (9–28)	3	16 (9–28)	0		
Antivenom	133	83 (62.4%)	50	0 (0%)	76	76 (100%)	-
Time from SB to AV, hour	83	9:00 (5:22–20:40)	0	-	76	9:00 (5:26–21:00)	-
AV 3 to 6 vials	83	76 (91.6%)	0	-	76	76 (100%)	-
Prescribed dose	83	6 (6–6)	0	-	76	6 (6–6)	-
Received dose	83	6 (4–6)	0	-	76	6 (6–6)	-
Early adverse reaction	83	17 (20.5%)	0	-	76	10 (13.2%)	-
Surgery	133	35 (26.3%)	50	15 (30%)	76	19 (25%)	0.536
Time from SB to surgery, days	35	7 (6–9)	15	8 (6 - 10)	19	7 (5–9)	0.137
Necrosectomy	35	16 (46%)	15	7 (47%)	19	9 (47%)	0.968
Infection	133	43 (32.3%)	50	19 (38%)	76	23 (30.3%)	0.367
Abscess	43	28 (65.1%)	19	11 (57.9%)	23	16 (69.6%)	0.432
Necrotizing fasciitis	43	8 (18.6%)	19	3 (15.8%)	23	5 (21.7%)	0.625
Cellulitis	43	12 (27.9%)	19	6 (31.6%)	23	6 (26.1%)	0.695
Time from SB to infection, days	42	6 (3–8)	19	5 (3–9)	22	6 (3–7)	0.344
Outcome							
ICU Length of Stay, days	128	3 (3–5)	50	4 (3–7)	71	3 (3–4)	0.009
Hospital LOS, days	133	10 (6–13)	50	11 (7–20)	76	9 (6–13)	0.002
Survival	133	131 (98.5%)	50	48 (96%)	76	76 (100%)	0.079

**Table 3 toxins-12-00662-t003:** Biological abnormalities recorded at admission and during hospital stay.

	**All Patients**		**Without Antivenom**		**With Antivenom**		***p***
**Parameter**	**Nbval**	**Result**	**Nb**	**Result**	**Nb**	**Result**	
Hemolysis	133	37 (27.8%)	50	14 (28%)	76	21 (27.6%)	0.964
Time from SB to end of hemolysis	128	18:28 (6:42–30:40)	47	22:00 (15:12–74:35)	74	16:30 (3:30–24:52)	0.002
Rhabdomyolysis	133	49 (36.8%)	50	15 (30%)	76	33 (43.4%)	0.129
Time from SB to normal CK	116	18:15 (7:21–28:07)	45	22:00 (13:00–63:30)	64	14:55 (5:25–22:32)	0.136
Hyperlactacidemia	132	27 (20.5%)	50	10 (20%)	75	14 (18.7%)	0.853
Coagulation							
Defibrinogenation	133	124 (93.2%)	50	42 (84%)	76	75 (98.7%)	0.002
Time from SB to normal fibrinogen	130	29:00 (22:18–45:55)	47	47:00 (28:30–96:13)	76	25:30 (20:42–32:45)	0.000
International Normalized Ratio (INR)	133	104 (78.2%)	50	32 (64%)	76	66 (86.8%)	0.003
Time from SB to normal INR	132	28:21 (17:55–54:54)	49	58:15 (27:30–91:26)	76	22:32 (17:00–35:03)	0.000
Partial thromboplastin time (PTT)	133	74 (55.6%)	50	23 (46%)	76	47 (61.8%)	0.080
Time from SB to normal PTT	132	18:20 (12:55–27:30)	49	25:00 (16:30–58:40)	76	16:05 (11:00–21:15)	0.000
Thrombocytopenia	133	52 (39.1%)	50	22 (44%)	76	29 (38.2%)	0.513
Time from SB to normal Platelet count	128	19:49 (5:37–107:23)	48	40:37 (12:47–156:34)	73	18:00 (3:03–62:00)	0.010
Platelet count, Giga/L	52	116 (75–130)	22	95 (73–121)	29	121 (99–132)	0.145
Decreased Factor II	133	48 (36.1%)	50	13 (26%)	76	34 (44.7%)	0.033
Time from SB to normal Factor II	110	17:29 (6:49–34:02)	34	24:55 (15:24–66:35)	69	15:10 (5:30–30:30)	0.000
Decreased Factor V	133	66 (49.6%)	50	13 (26%)	76	49 (64.5%)	0.000
Time from SB to normal Factor V	108	23:30 (15:05–37:34)	38	31:42 (19:00–65:52)	64	19:42 (11:00–30:30)	0.000
Decreased Factor VII	133	45 (33.8%)	50	11 (22%)	76	31 (40.8%)	0.029
Time from SB to normal Factor VII	95	15:10 (5:05–38:05)	33	22:25 (12:11–78:10)	56	8:00 (3:23–20:02)	0.000
Decreased Factor VIII	133	14 (10.5%)	50	4 (8%)	76	9 (11.8%)	0.488
Time from SB to normal Factor VIII	120	13:22 (6:07–22:52)	38	21:30 (14:00–57:56)	75	10:20 (5:05–18:48)	0.000
Decreased Factor IX	133	4 (3%)	50	0 (0%)	76	4 (5.3%)	0.099
Time from SB to normal Factor IX	123	12:20 (5:22–22:21)	40	20:35 (10:52–40:28)	76	9:50 (3:51–18:42)	0,000
Decreased Factor X	133	19 (14.3%)	50	8 (16%)	76	11 (14.5%)	0.815
Time from SB to normal Factor X	114	14:05 (5:32–25:27)	35	22:00 (12:35–81:20)	72	10:27 (3:51–20:32)	0.000
Decreased Factor XI	133	7 (5.3%)	50	0 (0%)	76	6 (7.9%)	0.042
Time from SB to normal Factor XI	122	12:42 (5:56–22:57)	40	20:35 (10:52–40:28)	75	10:00 (3:51–19:15)	0.000
Decreased Factor XII	133	13 (9.8%)	50	3 (6%)	76	9 (11.8%)	0.274
Time from SB to normal Factor XII	113	12:50 (5:40–23:00)	39	21:00 (11:35–39:37)	68	10:00 (3:41–19:07)	0.000

CK: Creatine Kinase.

**Table 4 toxins-12-00662-t004:** Biological abnormalities recorded at admission and during hospital stay in patients receiving 3 or 6 vials AV.

	With Antivenom		3 Vials of Antivenom		6 Vials of Antivenom		*p*
Parameter	Nb	Result	Nb	Result	Nb	Result	
Time from SB to end of hemolysis	74	16:30 (3:30–24:52)	14	20:00 (11:45–29:35)	57	11:00 (2:46–23:30)	0.048
Rhabdomyolysis	76	33 (43.4%)	14	6 (42.9%)	59	27 (46%)	0.844
Time from SB to normal CPK	64	14:55 (5:25–22:32)	13	18:50 (11:00–41:02)	48	13:20 (5:10–20:02)	0.021
Hyperlactacidemia	75	14 (18.7%)	14	1 (7.1%)	58	13 (22%)	0.195
Coagulation							
Defibrinogenation	76	75 (98.7%)	14	14 (100%)	59	58 (98%)	0.624
Time from SB to normal fibrinogen	76	25:30 (20:42–32:45)	14	33:30 (25:10–44:30)	59	25:10 (20:33–31:00)	0.064
International Normalized Ratio	76	66 (86.8%)	14	12 (85.7%)	59	52 (88%)	0.804
Time from SB to normal INR	76	22:32 (17:00–35:03)	14	31:15 (24:22–43:37)	59	20:50 (17:00–33:50)	0.227
Partial thromboplastin time (PTT)	76	47 (61.8%)	14	9 (64.3%)	59	36 (61%)	0.821
Time from SB to normal PTT	76	16:05 (11:00–21:15)	14	18:05 (16:02–28:22)	59	16:00 (11:00–20:15)	0.187
Thrombocytopenia	76	29 (38.2%)	14	7 (50%)	59	22 (37%)	0.382
Time from SB to normal Platelet count	73	18:00 (3:03–62:00)	13	22:00 (11:00–62:00)	57	18:00 (2:46–64:00)	0.904
Platelet count, Giga/L	29	121 (99–132)	7	129 (67–133)	22	121 (101–131)	0.570
Decreased Factor II	76	34 (44.7%)	14	8 (57.1%)	59	25 (42%)	0.318
Time from SB to normal Factor II	69	15:10 (5:30–30:30)	11	18:00 (13:35–32:30)	56	13:52 (5:07–27:55)	0.492
Decreased Factor V	76	49 (64.5%)	14	9 (64.3%)	59	37 (63%)	0.913
Time from SB to normal Factor V	64	19:42 (11:00–30:30)	12	25:40 (16:07–29:37)	50	17:50 (10:13–30:20)	0.306
Decreased Factor VII	76	31 (40.8%)	14	5 (35.7%)	59	25 (42%)	0.649
Time from SB to normal Factor VII	56	8:00 (3:23–20:02)	11	18:00 (10:40–24:15)	43	5:40 (2:23–19:25)	0.451
Decreased Factor VIII	76	9 (11.8%)	14	3 (21.4%)	59	5 (8%)	0.163
Time from SB to normal Factor VIII	75	10:20 (5:05–18:48)	14	18:00 (11:00–25:15)	59	9:00 (3:51–18:10)	0.008
Decreased Factor IX	76	4 (5.3%)	14	0 (0%)	59	4 (7%)	0.316
Time from SB to normal Factor IX	76	9:50 (3:51–18:42)	14	18:00 (9:30–25:15)	59	9:00 (3:40–18:10)	0.024
Decreased Factor X	76	11 (14.5%)	14	2 (14.3%)	59	9 (15%)	0.927
Time from SB to normal Factor X	72	10:27 (3:51–20:32)	14	18:00 (11:20–25:15)	55	7:25 (3:16–19:40)	0.074
Decreased Factor XI	76	6 (7.9%)	14	1 (7.1%)	59	4 (7%)	0.961
Time from SB to normal Factor XI	75	10:00 (3:51–19:15)	14	18:00 (11:00–25:15)	59	9:25 (3:01–18:28)	0.020
Decreased Factor XII	76	9 (11.8%)	14	3 (21.4%)	59	6 (10%)	0.249
Time from SB to normal Factor XII	68	10:00 (3:41–19:07)	12	19:15 (10:30–25:20)	53	7:00 (3:00–18:20)	0.021

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
