# Peer review of "Clinical Features and Management of Snakebite Envenoming in French Guiana"

_toxins, 2020, doi:10.3390/toxins12100662_

Round 1
Reviewer 1 Report
The authors summarized the effects of specific antitoxin treatments on viper bites at French Guiana from the results collected over a three-year period. As a result, side reactions due to administration of horse-derived antitoxin preparations appear at a very high rate, but the effectiveness of antitoxin administration such as shortening the length of hospital stay for treatment and shortening the time to normalize blood factors is generally shown. I think it is a valuable paper.
Please consider that by modifying the following points in the introduction of the main text, the purpose will become clearer and it will help the reader to understand this paper.
â‘ Describe what kind of viper inhabits French Guiana, that is, what kind of venomous snake bite occurs.
(2) There is no description about the cause of death of a 39-year-old man who died in 2017. A venomous snake can be estimated from the symptoms of death.
(3) Please describe the reason for selecting the antitoxin “Antivipmyn Tri” used in this paper as a therapeutic antitoxin. Although it is described in detail in the discussion, the intention of the text will be easier to understand if it is briefly stated in the introduction.
Author Response
(1) We added : More than 99 snake species have been identified in French Guiana, of which 12 are venomous, and five of these (Bothrops atrox, Bothrops brazili, Bothrops bilineatus, Lachesis muta and Micruruss sp) are responsible for most cases of envenomation [4].
(2) We added: In January 2017, a 39-year-old man died from severe hemorrhagic complication after being bitten by a snake,
(3) We added : Antivipmyn Tri® was the sole antivenom against the genus Bothrops authorized for use by the French authorities and available in the national antivenom bank.
Reviewer 2 Report
The paper describes a study with patients bitten by snakes attended in a UCI facility in French Guiana. Some patients have received antivenom, and others have not, because of a supply disruption of antivenom. Results clearly demonstrate that antivenom for treating snakebites significantly reduced normalizing hemostatic parameters, even with the late time in which the antivenom was administered and with different levels of some adverse reactions.
The results were well discussed, and the data support the conclusion.
I think the authors should show, in lane 113, that the two patients who died from systemic hemorrhage during the study did not receive antivenom.
Author Response
We added: Two patients died from systemic hemorrhage during the study period. Both of them did not received antivenom.